# Individual differences among deep neural network models

Johannes Mehrer [1✉], Courtney J. Spoerer[1], Nikolaus Kriegeskorte [2] & Tim C. Kietzmann [1,3✉]

Deep neural networks (DNNs) excel at visual recognition tasks and are increasingly used as a modeling framework for neural computations in the primate brain. Just like individual brains, each DNN has a unique connectivity and representational profile. Here, we investigate individual differences among DNN instances that arise from varying only the random initialization of the network weights. Using tools typically employed in systems neuroscience, we show that this minimal change in initial conditions prior to training leads to substantial differences in intermediate and higher-level network representations despite similar network-level classification performance. We locate the origins of the effects in an under-constrained alignment of category exemplars, rather than misaligned category centroids. These results call into question the common practice of using single networks to derive insights into neural information processing and rather suggest that computational neuroscientists working with DNNs may need to base their inferences on groups of multiple network instances.

---

[1] MRC Cognition and Brain Sciences Unit, University of Cambridge, 15 Chaucer Road, Cambridge CB2 7EF, UK. [2] Zuckerman Institute, Columbia University, 3227 Broadway, New York, NY 10027, USA. [3] Donders Institute for Brain, Cognition and Behaviour, Radboud University, Montessorilaan 3, 6525 HR Nijmegen, Netherlands. ✉email: johannes.mehrer@mrc-cbu.cam.ac.uk; t.kietzmann@donders.ru.nl

  1

Deep neural networks (DNNs) have recently moved into the focus of the computational neuroscience community. Having revolutionized computer vision with unprecedented task performance, the corresponding networks were soon tested for their ability to explain information processing in the brain. To date, task-optimized deep neural networks constitute the best model class for predicting activity across multiple regions of the primate visual cortex[1–5]. Yet, the advent of computer vision models in computational neuroscience raises the question in how far network-internal representations generalize, or whether network instances, just like brains, exhibit individual differences due to their distinct connectivity profiles. Large differences would imply that the common practice of analyzing a single network instance is misguided and that groups of networks need to be analyzed to ensure the validity of insights gained.

Here we investigate individual differences among deep neural networks that arise from a minimal experimental intervention: changing the random seed of the network weights prior to training while keeping all other aspects identical. With this, we build on and expand previous investigations of network similarities in the machine learning community. Most notably, researchers have previously employed variants of linear canonical correlation analysis (CCA) and centered-kernel alignment (CKA) to compare network-internal representations. Using singular value decomposition as a pre-processing step before CCA, singular vector CCA (svCCA) was used to compare representations across networks[6]. The authors report diverging network solutions predominantly in intermediate network layers. Building on svCCA, projection-weighted-CCA (pwCCA) was introduced, which assigns different weights to CCA vectors according to their effect on the output vectors. Using this extension, the authors observed decreasing network similarities with increasing layer depth[7]. Finally, Kornblith et al. introduced centered-kernel alignment (CKA)[8], a neuroscience-inspired technique that builds upon previous CCA solutions. Using this analysis approach, the authors demonstrated that task-trained networks developed more similar representations than random networks, even when task training was performed on different object categorization data sets. CKA furthermore identified meaningful layer correspondence between networks trained from different network initializations. This effect was strongest in early and intermediate network layers, indicating diverging network representations in later layers.

To allow for direct links to the computational neuroscience community, we here rely on analysis techniques commonly used in the field. In particular, we employ representational similarity analysis (RSA)[9], a multivariate analysis technique from systems neuroscience that is typically used to compare computational models to brain data. RSA is based on the concept of representational dissimilarity matrices (RDMs), which characterize a system's inner stimulus representations in terms of pairwise response differences. Together, the set of all possible pairwise comparisons provides an estimate of the geometric arrangement of the stimuli in high-dimensional activation space[10].

Representations of two systems are considered similar if they emphasize the same distinctions among the stimuli, i.e., to the degree that their RDMs agree. Comparisons on the level of RDMs, which can be computed in source spaces of different origin and dimensionality, thereby side-step the problem of defining a correspondence mapping between units of observation (e.g., between fMRI voxels and network units). An explicit design feature of the current investigation is to use the same analysis approach for comparisons across network instances as commonly employed in neuroscience. This has the advantage that the set of results will be directly applicable to the common neuroscientific use case. To quantify RDM agreement across network instances,

we define representational consistency as the shared variance between network RDMs (squared Pearson correlation of the upper triangle of the RDMs; Fig. 1, see "Methods" for details, pseudocode provided as Supplementary Information).

Based on this analysis approach, we visualize the internal network representations and test them for consistency (see below for an analysis pipeline overview). We find significant individual differences among deep neural network instances that vary only in the initial random seed. This finding replicated across different network architectures, training sets, and distance measures. The size of the effect is shown to be comparable to differences arising from training networks with different input statistics. Subsequently, we explore possible causes for these individual differences and find that they largely originate from a varying alignment between category exemplars rather than category centroids. We then investigate the effects of network regularization via Bernoulli dropout at training and test time. While it can increase representational consistency, it does not overcome network individual differences. Finally, we analyze and visualize the development of individual differences across network training trajectories. We find a strong negative relationship between task performance and representational consistency, indicating that task training increases individual network differences.

## Results

**Individual differences emerge in deeper network layers**. We here investigate the extent to which deep neural networks exhibit individual differences. We approach this question by training multiple instances of the All-CNN-C network architecture[11] and a custom architecture (All-CNN-7) on an object classification task (CIFAR-10[12]), followed by an in-depth analysis of resulting network-internal representations. Network instances varied only in the initial random assignment of weights, while all other aspects of network training were kept identical. All networks performed similarly in terms of classification accuracy (ranging between 84.4–85.9% and 77.6–78.95% top-1 accuracy for All-CNN-C, and All-CNN-7, respectively).

To study and compare network-internal representations, we extracted network activation patterns for 1000 test images (100 for each of the CIFAR-10 categories, Fig. 1a) and characterized the underlying representations in terms of pairwise distances in the high-dimensional activation space (Fig. 1b). The reasoning of this approach is that if two images are processed similarly in a given layer, then the distance between their activation vectors will be low, whereas images that elicit distinct patterns will have a large activation distance. The matrix of all pairwise distances (size $1000 \times 1000$) thereby describes the representational geometry of the test images, i.e., how exemplars of various object categories are grouped and separated by the units of a given layer (see below for a detailed depiction of the RDM structure).

To visualize the representational geometries of different network instances and layers, we projected the data into 2D using multidimensional scaling (MDS, metric stress). As can be seen in Fig. 2 for two exemplary cases of All-CNN-C, subsequent network layers increasingly separate out the different image categories, in line with the training objective (see Supplementary Fig. 1 for point-wise stress estimates).

Moving closer to the question of individual differences in network representations, we next compared the arrangement of activation vectors across network layers and instances (2nd level RSA, see "Methods" section). That is, we again computed pairwise distances, but this time not based on original activation patterns, but rather based on the extracted network RDMs. This 2nd level comparison has multiple benefits. For one, focus on pattern distances offers a characterization of network-internal

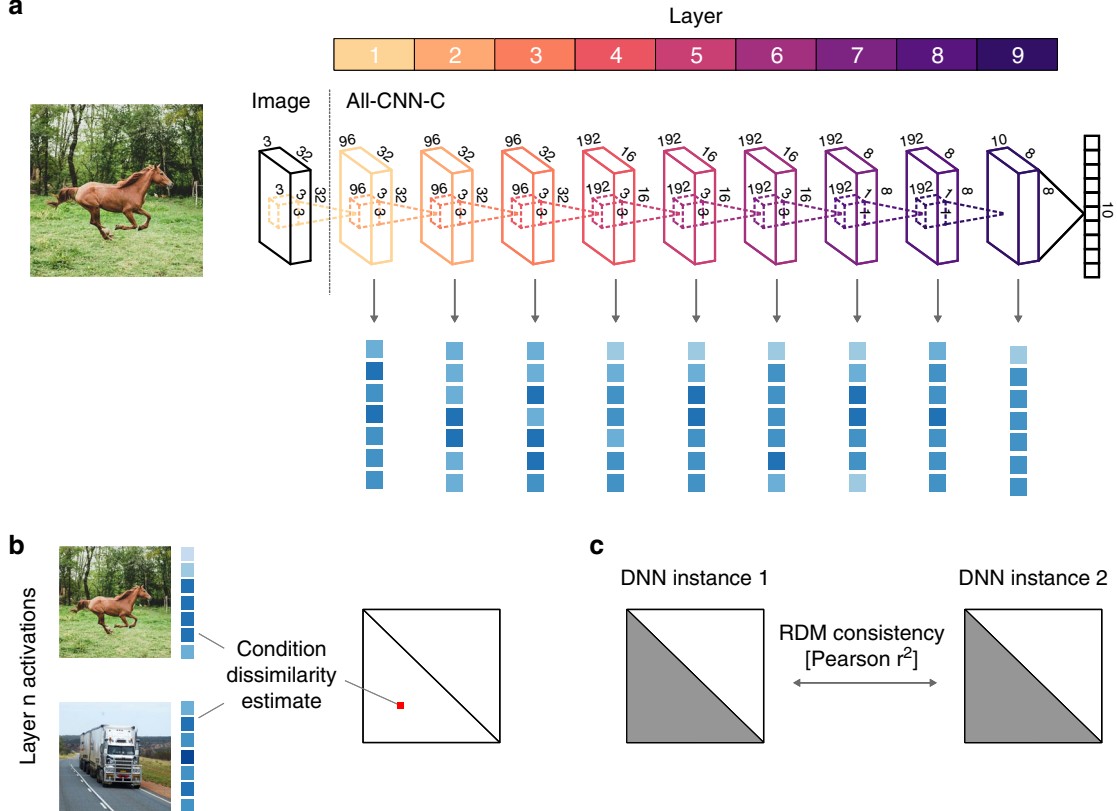

**Fig. 1 Comparing network-internal representations using RSA and representational consistency. a** Our comparisons of network-internal representations were based on their multivariate activation patterns, extracted from each layer of each network instance as it responded to each of 1000 test images. **b** These high-dimensional activation vectors were then used to perform a representational similarity analysis (RSA). The fundamental building blocks of RSA are representational dissimilarity matrices (RDMs), which store all pairwise distances between the network's responses to the set of test stimuli. Each test image elicits a multivariate population response in each of the network's layers, which corresponds to a point in the respective high-dimensional activation space. The geometry of these points, captured in the RDM, provides insight into the nature of the representation, as it indicates which stimuli are grouped together, and which are separated. **c** To compare pairs of network instances, we compute their representational consistency, defined as the shared variance between network RDMs.

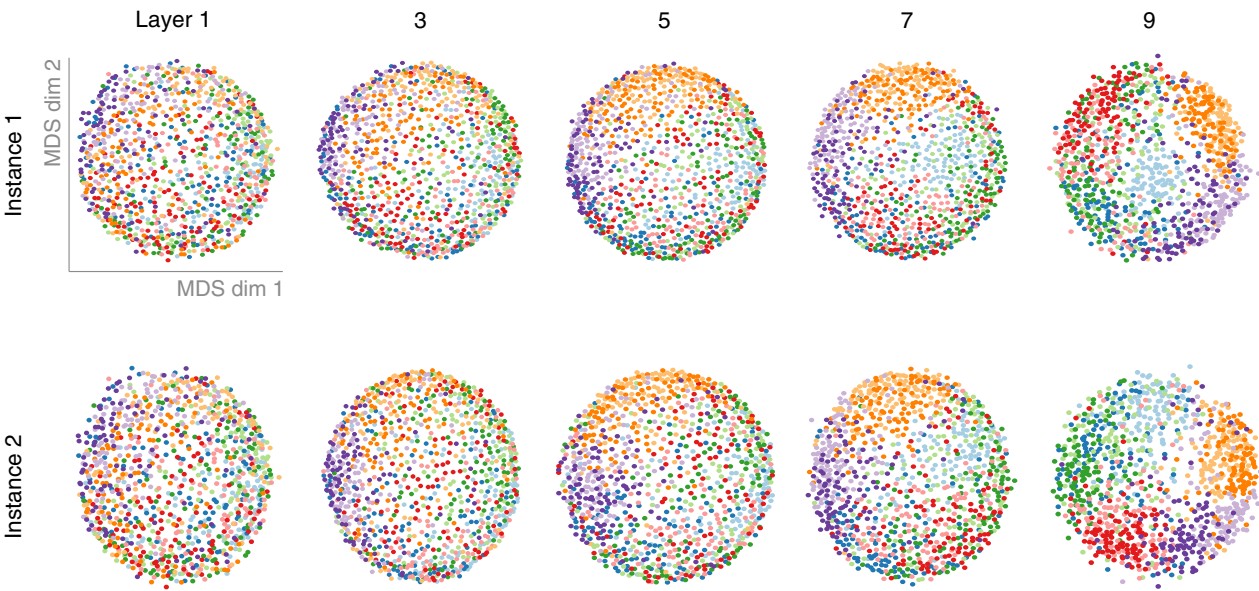

**Fig. 2 Representational geometries at different network depths of two DNN instances.** The internal representations of two network instances were characterized based on their representational geometries. We computed the pairwise distances (correlation distance) between activity patterns in response to 1000 test stimuli from 10 visual categories and visualized them in 2D via multidimensional scaling (MDS; metric stress criterion; categories shown in different colors). With increasing depth, networks exhibit increased category clustering.

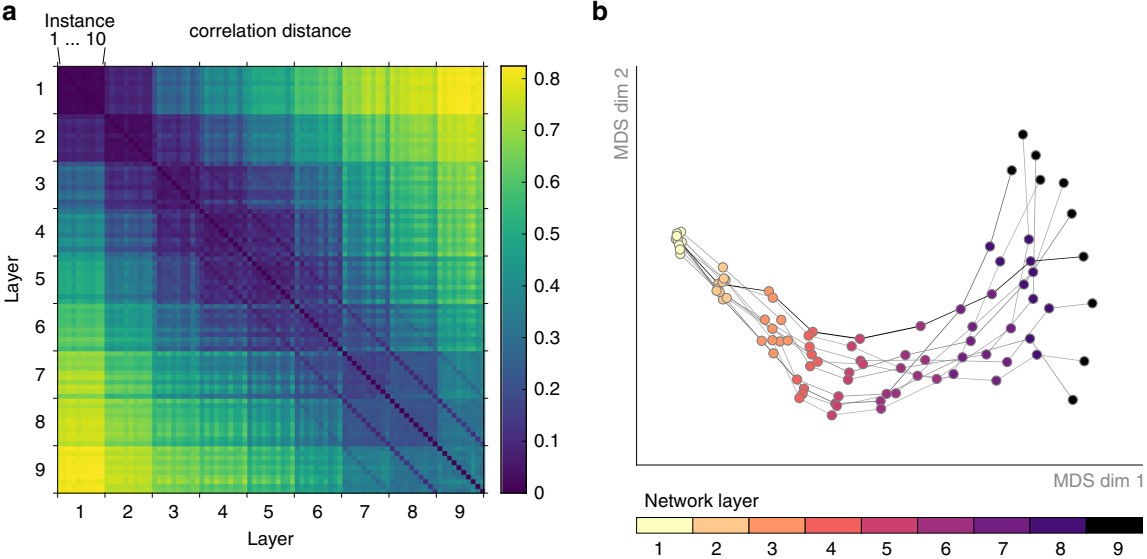

**Fig. 3 Network individual differences emerge with increasing network depth.** **a** We compare the representational geometries across all network instances (10) and layers (9 convolutional) for All-CNN-C by computing all pairwise distances between the corresponding RDMs. **b** We projected the data points in **a** (one for each layer and instance) into 2D via MDS. Layers of individual network instances are connected via gray lines. While early representational geometries are highly similar, individual differences emerge gradually with increasing network depth.

representations that is largely invariant to rotations of the underlying high-dimensional space, including a random shuffle of network units (see Supporting Information for more details). Secondly, representational spaces of varying dimensionality can be directly compared, as the dimensionality of the RDM is fixed by the number of test images used.

We computed this second-level distance measure (i.e., the dissimilarity between RDMs rather than activation vectors) for all network layers and instances. Visualizing the respective distances in 2D (MDS, metric stress), we observe that representations diverge substantially with increasing network depth (Fig. 3). While different network instances are highly similar in layer 1, indicating agreement in the underlying representations, subsequent layers diverge gradually with increasing network depth. Note that for later layers, the blue stripes parallel to the main diagonal indicate higher similarity across layers within a given network instance compared to the similarities across instances for a given network layer (Supplementary Fig. 2).

**Representational consistency: quantifying DNN differences.** Following this initial qualitative assessment, we performed quantitative analyses for each network layer by testing how well the distribution of representational distances generalizes across network instances. This was accomplished by computing representational consistency, defined as the shared variance between the upper triangle of the respective RDMs (Fig. 1c, each triangle contains 499,500 distance estimates, results are obtained from 45 pairwise network comparisons for each respective layer and network architecture as 10 network instances are trained for each architecture, analysis pseudocode provided in the "Methods" section). This measure of consistency is based on all pairwise distances between category exemplars (100 exemplars for 10 categories each). We, therefore, refer to this as exemplar-based consistency (see "Methods" section for further details).

Representational consistency is based on comparing network RDMs. To compute these RDMs, we used correlation distance as a dissimilarity measure, as it is currently the most frequently used distance measure in systems and computational neuroscience (later sections will investigate further distance measures).

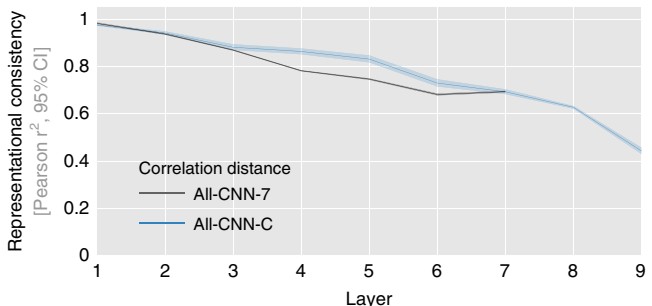

**Fig. 4 Representational consistency decreases with increasing network depth.** Average representational consistency for each network layer computed across all pairwise comparisons of network instances (45 comparisons for 10 instances, computed separately for two network architectures). Error bars indicate 95% confidence intervals (CI, bootstrapped).

shown in Fig. 4, representational consistency drops substantially with increasing network depth for both network architectures. For All-CNN-C, consistency (i.e., shared variance in distance estimates), drops to 44%, for All-CNN-7, consistency drops to 71%.

To get better insights into the size of this effect, additional networks were trained (i) based on different images originating from the same categories, and (ii) based on different categories (see "Methods" section for details). The observed drops in consistency for different weight initializations are comparable to training the networks with the same distribution of categories but completely separate image data sets (Supplementary Fig. 3, blue vs. orange).

To ensure that the effects observed are not specific to correlation distance used in computing the RDMs, additional analyses were performed based on cosine, (unit length pattern-based) Euclidean distance and norm difference (measuring the absolute difference in the norm activation vectors, Fig. 5). In all cases, representational consistency was observed to drop

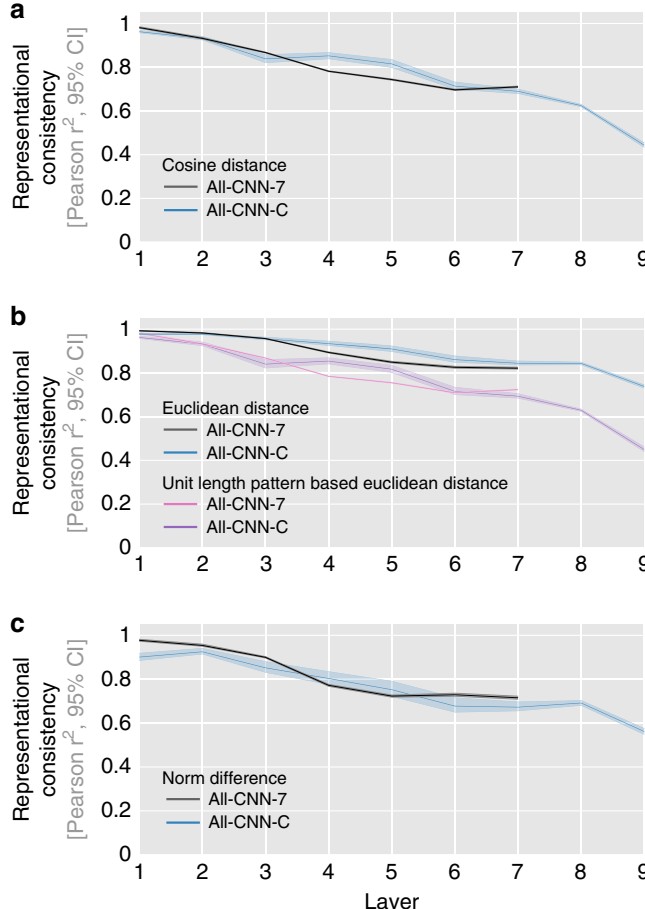

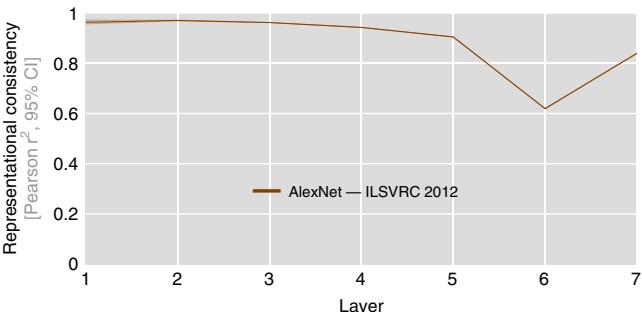

**Fig. 6 Representational consistency decreases in AlexNet.** We repeated our above analyses of representational consistency on a set of AlexNet instances trained on large-scale object classification data set ILSVRC 2012. Again, we only vary the initial random seed of the network weights. In line with our previous results, we observe a decrease in representational consistency from early to late network layers. The minimal average consistency is observed in layer fc6, which exhibits 62% of the shared variance across network RDMs. As AlexNet requires the input of size 224 × 224, which is significantly larger than the 32 × 32 image size of CIFAR-10 used earlier, we created an independent set of larger images from the same 10 categories while following the same data set structure (100 images per CIFAR-10 category). Ten network instances correspond to 45 pairwise distance estimates per network layer, average representational consistency shown here with 95% confidence intervals (bootstrapped).

**Fig. 5 Representational consistency decreases irrespective of distance measure.** Representational consistency decreases with increasing layer depth for both tested DNN architectures, and across multiple ways to measure distances in multivariate population responses (cosine (**a**), Euclidean distance and unit length pattern-based Euclidean distance (**b**), and differences in vector norm (**c**)). Average representational consistency shown for each layer, computed across all pairwise comparisons of network instances (45 comparisons for 10 instances), together with a 95% bootstrapped confidence interval.

considerably with increasing network depth. These results demonstrate that while different network instances reach very similar classification performance, they do so via distinct internal representations in the intermediate and higher network layers.

The above results represent an important existence proof for substantial DNN individual differences that can occur in computational neuroscience analysis pipelines. To expand our experiments to network architectures commonly used to predict brain data[3,4,13–15], we trained and tested 10 network instances of a recent version of AlexNet[16] on a large-scale object classification data set ILSVRC 2012[17]. As AlexNet requires larger input images than the previously used CIFAR-10 (width/height of 224px vs. 32px), we sampled a new test set that nevertheless reflects the categorical structure of CIFAR-10: 100 images from each of the 10 CIFAR-10 classes were used to compute network RDMs. Replicating our previous results, consistency was also found to decrease with increasing network depth for AlexNet. The strongest individual differences were observed in fully connected layer fc6 (62% explained variance). We observe consistency levels of 84% in the penultimate layer (Fig. 6).

**Causes of decreasing representational consistency.** We have demonstrated above that different network instances can exhibit substantial individual differences in their internal representations. Next, we investigate potential mechanisms that may contribute to this effect.

Our first analyses are based on the hypothesis that the training goal of maximal category separability does not put a strong constraint on the relative positions of categories and category exemplars in high-dimensional activation space. To investigate this possibility, for the 10 network instances of All-CNN-C used in the previous section, we computed a category clustering index (CCI) for each network layer using the network responses to the set of 1000 test images (drawn from 10 categories). CCI is defined as the normalized difference in average distances for stimulus pairs from different categories (across) and stimulus pairs from the same category (within): CCI = (across − within)/(across + within) (see "Methods" section). CCI can be regarded as a multivariate extension to a previously introduced category tuning index[18]. It approaches zero with no categorical organization and is positive if stimuli from the same category cluster together (maximum possible CCI = 1). We find a negative relationship between CCI and representational consistency (Pearson $r = -0.92$, $p = 0.001$; robust correlation[19], see Supplementary Fig. 4). This indicates that network layers that separate categories better exhibit stronger individual differences, as measured via nonlinear representational consistency. These results are in line with previous findings demonstrating that linear class-separability increases with network depth[20], and observations of decreasing network similarities with increasing layer depth[6–8,21].

A negative relationship between category clustering and representational consistency is compatible with two possible scenarios: first, networks could exhibit a different arrangement of the overall category clusters. While linear class-separability in the penultimate network layer is required for successful task completion, this does not necessarily imply centroid consistency. That is, we cannot exclude a scenario in which a pair of networks

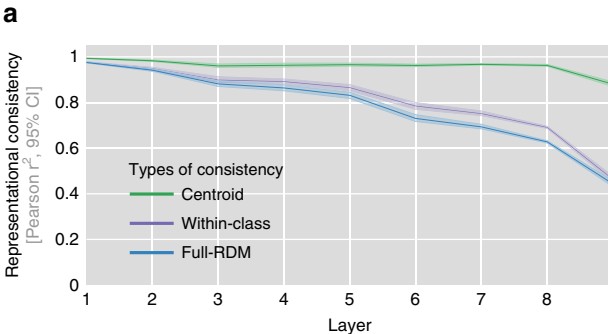
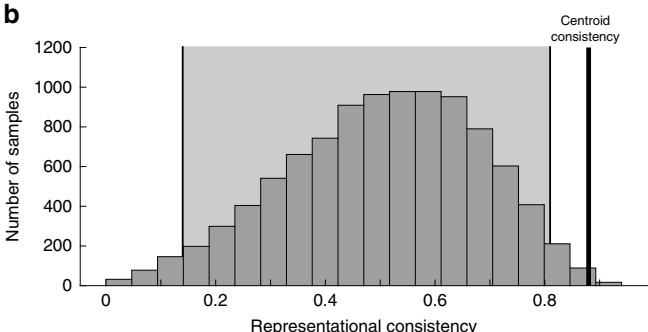

**Fig. 7 Category centroids are highly consistent across network instances. a** Centroid-based representational consistency (green) remains comparably high throughout, whereas the consistency of within-category distances decreases significantly with increasing network depth (error bars indicate 95% confidence intervals, average data shown, computed from 45 network comparisons across 10 network instances). This indicates that differences in the arrangement of individual category exemplars, rather than large-scale differences between class centroids are the main contributor to the observed individual differences. **b** High centroid-based representational consistency cannot be explained by the smaller RDMs or the averaging of multiple response patterns, as centroids of randomly sampled classes show a significantly lower mean consistency (95% CI in the light gray background).

show a similar level of class-separability, albeit a different overall arrangement of class centroids. In this case, class-separability would be high in both cases, but centroid consistency would be low. Second, focusing on distances within-category clusters, different arrangements of individual exemplars within the clusters could be the source of individual differences. Both, overall category and category exemplar placement are not constrained by the categorization training objective.

To investigate the variability in general cluster placement, we computed representational consistency based on the ten category centroids (RDMs computed from the pairwise distances of average response patterns for each category). This analysis revealed that centroid consistency is considerably higher than the previous exemplar-based consistency (Fig. 7a, $\mu_{\text{centroid-based}} = 0.8801$, $\text{CI}_{95} = [0.8700, 0.8905]$ vs. $\mu_{\text{exemplar-based}} = 0.4429$, $\text{CI}_{95} = [0.4291, 0.4551]$ for correlation distance; $\mu_{\text{centroid-based}} = 0.9515$, $\text{CI}_{95} = [0.9450, 0.9571]$ vs. $\mu_{\text{exemplar\_based}} = 0.7384$, $\text{CI}_{95} = [0.7312, 0.7466]$ for Euclidean distance, all computed for the final layer of All-CNN-C). This finding cannot be explained by the lower number of pairwise comparisons (45 vs. 499,500 for centroid and stimulus RDMs, respectively) or the operation of averaging large numbers of activation patterns (each centroid is computed based on 100 activation patterns), as computing centroids from random stimulus assignments yielded significantly lower centroid-based representational consistency (95% CI of centroid-based consistency based on random class assignment [0.14, 0.81], Fig. 7b). Together, these results suggest that category centroids are located in similar geometric arrangements in-network instances trained off of different seeds, rendering overall category placement a less likely source of the observed individual differences.

The reliable arrangement of category centroids suggests that a main source of the observed individual differences lies in the arrangement of category exemplars within the category clusters themselves. This view was corroborated by computing consistency, not on the whole exemplar-based RDM that contains all pairwise distances, but only on the dissimilarities of exemplars of the same categories (within-category consistency, see "Methods" section). Focusing on within-category distances, we observe a drop in consistency that is largely comparable to the original decrease for exemplar-based consistency computed based on the whole RDM (Fig. 7a).

In addition to an individual placement of category centroids and category exemplars, properties of the used dissimilarity measures could be a source for lower representational

consistency, especially in cases of a rotated representational space. Many commonly used DNNs use rectified linear units (ReLUs) as a nonlinear operation, resulting in unit activations ≥ 0. If different network instances learned different projections that are equivalent to a rotation in this all-positive space, then this change will not affect classification performance. However, it can affect estimates of correlation and cosine distances (Supplementary Fig. 5). As shown in Supplementary Fig. 6, rotations around the origin have additional effects on correlation distances but not on cosine distances.

To test the magnitude of this effect, we subtracted the mean activation pattern across all test images from the units of a given layer (cocktail-blank normalization). This normalization led to increases in representational consistency for RDMs computed using correlation or cosine distance (see Supplementary Fig. 7 for details). While the size of the effect is comparably small, these results indicate that a cocktail-blank normalization can be of potential benefit when comparing correlation- or cosine-based RDMs of multiple DNNs or DNNs and brain data.

**Bernoulli dropout affects representational consistency.** An explanation of individual differences via missing constraints imposed by the training objective raises the possibility that explicit regularization during network training can provide the missing representational constraints[22,23]. We investigated this possibility experimentally by training networks at various levels of Bernoulli dropout. We trained 10 network instances of All-CNN-C for each of 9 dropout levels (Bernoulli dropout probability ranging from 0 to 0.8, a total of 90 network instances trained) and subsequently tested the resulting representations for their ability to classify input as well as for their representational consistency. To test for differences in task performance, we computed the top-1 categorization accuracy for the training- and test data. For the test data, we compare network performance with and without dropout at the time of inference. In line with the literature[23], we find reduced training accuracy, but enhanced test accuracy at moderate dropout levels (Fig. 8a).

The effects of dropout training on representational consistency were investigated using layer 9 of All-CNN-C, which exhibited the lowest consistency levels in our original analyses. Focusing on the effects of using Bernoulli dropout during training, we observe that it reduced network individual differences. The highest consistency was found for a dropout probability of 0.6, which led to an average of 64.7% shared variance (rightmost

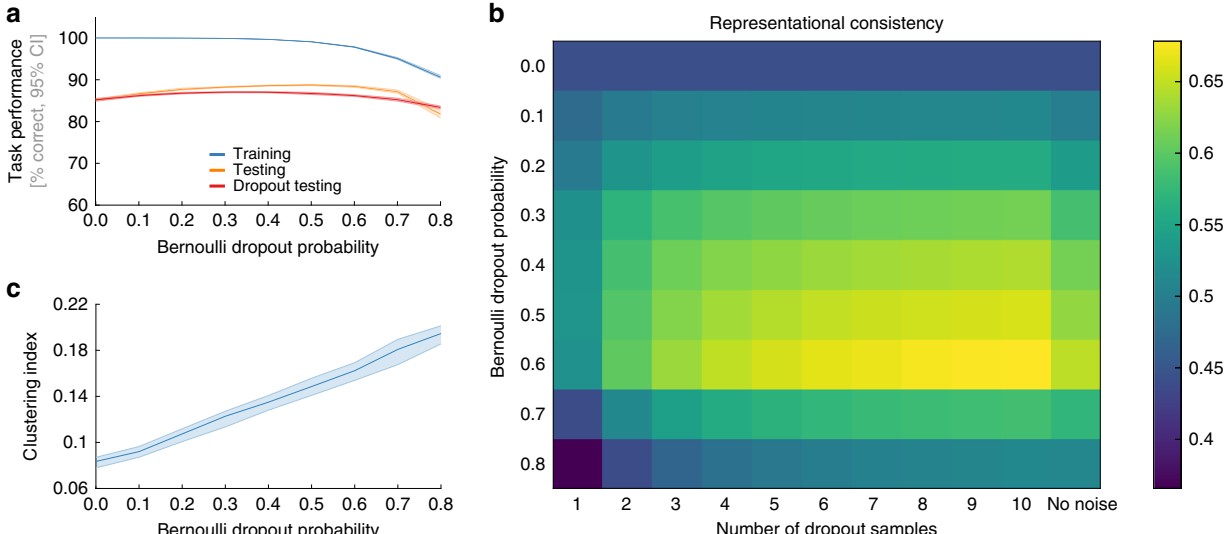

**Fig. 8 Effects of Bernoulli dropout on task performance and representational consistency. a** Task performance, the average across all 10 network instances shown with 95% CI for the training set (blue), test set (orange), and when using dropout sampling at inference time for the test set (red, 1 sample). **b** Average representational consistency in the final convolutional layer of All-CNN-C as a function of dropout probability during training and test (dropout probability at test time set to equal dropout probability during training, consistency derived from 45 network pairs). When using dropout at test time, multiple samples can be drawn for each stimulus in the test set (creating multiple RDMs). Consistency for network pairs was computed for the respective average RDM for each instance. Consistency was observed to be highest when 10 samples were obtained from a DNN trained and tested at a dropout rate of 60%. **c** The clustering index for the penultimate layer of All-CNN-C increases with increasing Bernoulli dropout probability (10 network instances, error bars 95% CI).

column in Fig. 8b)—a marked increase compared to the 44% observed without dropout.

In analogy to our analyses of test accuracy in which we apply Bernoulli dropout at the time of inference, we investigated how far obtaining multiple test samples of the activation patterns affect representational consistency. For each network instance, we computed 10 RDM samples while keeping the dropout mask identical across network instances and the dropout rate identical to training. Like this, we obtained 10 RDM samples for each network instance and subsequently use the average RDM to compute representational consistency (see "Methods" section). We find that increasing the number of RDM samples led to increased representational consistency for all dropout levels (Fig. 8b). Maximum representational consistency was observed for 10 RDM samples at a dropout probability of 0.6, reaching an average of 67.8% shared variance. This suggests that dropout applied during training and test can increase the consistency of the representational distances across network instances.

To investigate a possible mechanism of how dropout may have positively affected consistency, we computed the category clustering index, as previously defined, for the penultimate layer of All-CNN-C trained at various dropout levels. The reasoning for this was that if category centroids are highly consistent, then stronger clustering of category exemplars around the centroids will at the same time yield higher overall representational consistency. As shown in Fig. 8c, we observe a positive relationship between dropout probability and category clustering, supporting our hypothesis. However, while clustering is increased further for dropout levels >0.6, representational consistency starts to decrease. We explain this effect by observing that centroid consistency is significantly decreased ($\mu_{\mathrm{dropout}=0.8} = 0.7422$, CI$_{95}$ = [0.6881, 0.7854]) compared to the no dropout case ($\mu_{\mathrm{no\_dropout}} = 0.8801$, CI$_{95}$ = [0.8700, 0.8905]) for the highest dropout level of 0.8. Thus, while denser clustering around centroids increases consistency in cases where the centroids themselves are consistent

(here up to dropout levels of 0.6), high levels of dropout break the centroid consistency and therefore lead to an overall decrease in representational consistency.

**Representational consistency across training trajectories**. We observed above that representational consistency across network instances is remarkably stable for category centroids. This raises the question of whether this alignment is the result of task training, or whether category centroids are already well-aligned early during training. To investigate the effects of training, we computed representational consistency (exemplar-based and centroid-based) across network instances and training epochs for the final layer of All-CNN-C. These analyses indicate that networks exhibit high consistency after the first training epoch, which decreases from thereon (Fig. 9a, b). From about 50 epochs onwards, networks exhibit relatively stable representations with each network remaining on its own learning trajectory (Fig. 9a, multiple diagonal lines indicate stable representations across training compared to other network instances). These results indicate that task training increases individual differences, whereas learning trajectories of individual networks across time remain surprisingly robust. In line with this, representational consistency and task performance exhibit a strong negative relationship (Pearson $r = -0.91$, $p < 0.001$; robust correlation[19], Fig. 9b–d). In line with our earlier results, category centroids are significantly more consistent, even for the earliest epochs, but otherwise exhibit similar training effects (Supplementary Fig. 8).

## Discussion
In a series of experiments, we here investigated how the minimal intervention of changing the initial set of weights in feedforward deep neural networks, while keeping all else constant, affects their internal representations. In our analysis pipeline, we explicitly chose to match techniques commonly used in computational

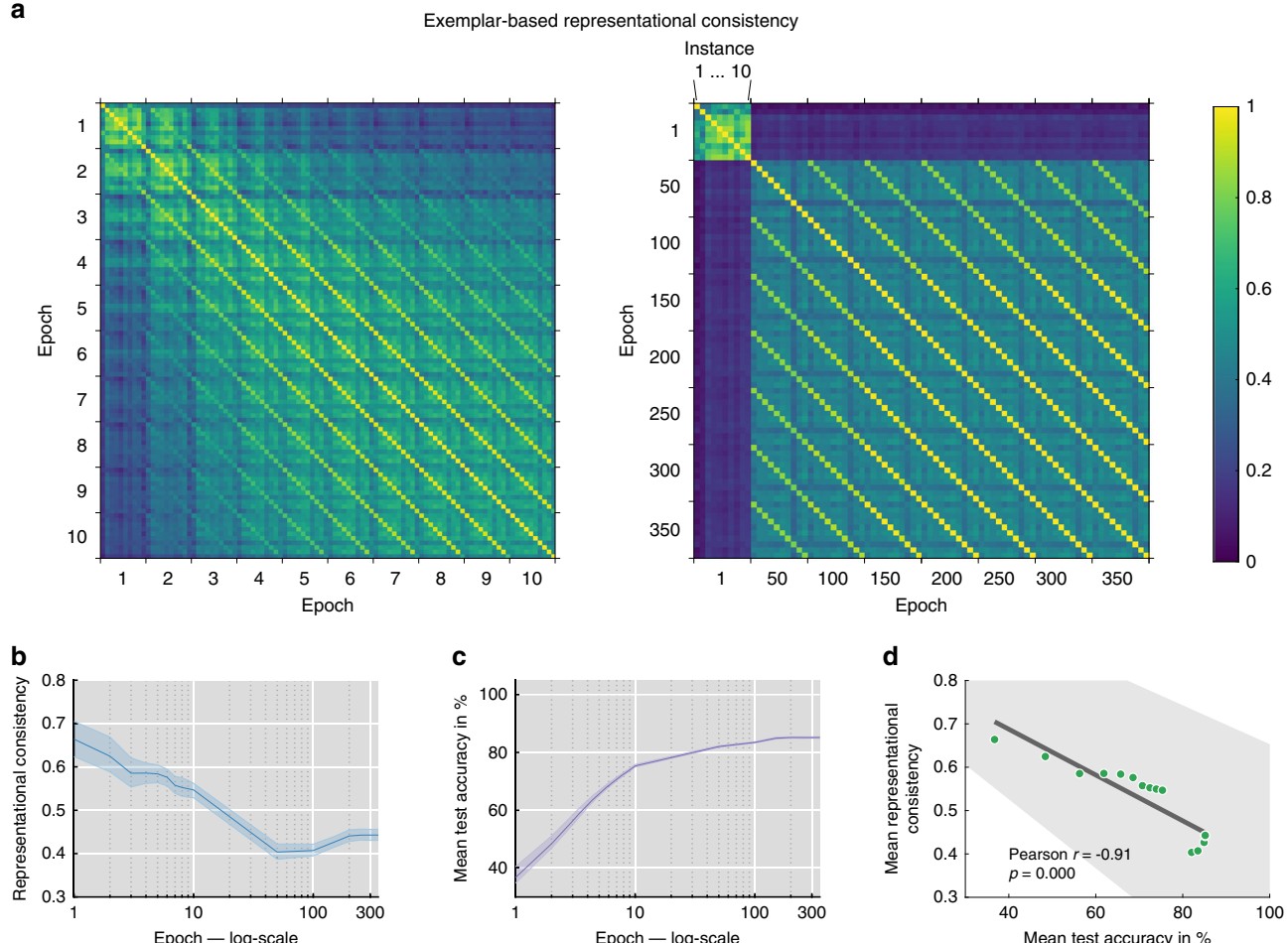

**Fig. 9 Final-layer representational consistency (exemplar-based) across training epochs. a** Comparing representational consistency across early epochs [1 to 10] (left) and throughout all training epochs [1 to 350 in steps of 50] (right). Lines parallel to the main diagonal indicate that network instances remain on their distinct representational trajectory compared to other networks. Average consistency shown across 45 network pairs, derived from 10 network instances. **b** Representational consistency, computed and averaged across all network pairs (45 pairs total) for each training epoch, demonstrates increasing individual differences with training (shown with 95% CI). **c** Test performance across training (average top-1 accuracy across 10 network instances with 95% CI). **d** Representational consistency and test performance exhibit a strong negative relationship (Pearson $r = -0.91$, $p < 0.001$; robust correlation) indicating that task training enhances individual differences (dots represent network training epochs, error bar indicates 95% CI).

neuroscience to compare DNNs to brain data. The current set of results therefore directly transfers to the relevant neuroscientific publications. Moreover, a focus on RSA allowed us to test the effects that different distance measures have on individual difference estimates. In addition to insights into the nature of these differences, this is of importance to the neuroscience community as different distance measures emphasize different aspects of the data that could be of specific interest in a given experimental setting. For example, the vector angle of neural population responses in IT was recently shown to encode different cognitive aspects than their magnitude[24]. This requires computational models to be probed in the same manner.

Operationalized as representational consistency, we demonstrated that significant individual differences emerge with increasing layer depth. This finding held true for different network architectures and various distance measures that are commonly used to compute the RDMs (correlation distance, cosine distance, variants of Euclidean distance, and norm differences). RDMs computed from Euclidean distances showed the least differences. This can be attributed in part to the fact that this distance measure is sensitive to differences in overall network activation magnitudes, which may overshadow more nuanced

pattern dissimilarities, in line with the lower consistency observed for norm-standardizing Euclidean distances (unit length pattern-based Euclidean distance). The observation of increased differences with increasing network depth is in line with findings from the domain of machine learning that compared network representations using methods related to CCA (svCCA[6], pwCCA[7], and CKA[8]). Although further experiments are required, we expect our results to generalize to representations learned by (unrolled) recurrent neural network architectures[25,26], if not explicitly constrained[27]. For an investigation of recurrent neural network dynamics arising from various network architectures, see Maheswaranathan et al.[28].

Having demonstrated significant network individual differences, we explored multiple non-exclusive explanations for the effects. Based on the hypothesis that the network training objective of optimizing for categorization performance may not sufficiently constrain the arrangement of categories and individual category exemplars, we analyzed category clustering, centroid arrangement, and within-category dissimilarities. These analyses demonstrate a high consistency of category centroids, rendering differences between individual category exemplars the main contributor to the differences observed. As an additional

source of variation, we identified an interaction between properties of the distance measures used and the ReLU nonlinearity commonly used in DNNs. We showed that cocktail-blank normalization in the DNN activation patterns can increase consistency for measures that are not robust to rotations that are not centered around zero (cosine distance) or general rotations (correlation distance).

In addition to these sources of variation, we showed that applying Bernoulli dropout during training and test can enhance representational consistency estimates. As a partial explanation for this increase, we demonstrated that dropout enhances category clustering around highly consistent category centroids. Consistent centroid positions furthermore imply that network alignment by reweighted network readout[3] will likely not enhance representational consistency.

Our finding of considerable individual differences has implications for computational neuroscience where single network instances are often used as models of information processing in the brain. If a given study compared only a single network instance to brain data, then it remains a possibility that the observation of a good (or bad) fit would be partially due to chance, as training a network off of a different random seed could have resulted in substantially different internal representations (and thereby in a different estimate of the alignment between the model and the brain). Neglecting the potentially large variability in network representations will therefore likely limit the generality of claims that can be derived from comparisons between DNNs and neural representations. The current set of results thereby marks an important existence proof for individual differences among network instances using neuroscientific analysis techniques. Although we here present multiple approaches that can increase consistency (cocktail-blank, dropout, and the choice of distance measure), significant differences remain. For computational neuroscience to take full advantage of the deep learning framework[29–32], we, therefore, suggest that DNNs should be treated similarly to experimental participants and that analyses should be based on groups of network instances.

Representational consistency as defined here will give researchers a way to estimate the expected network variability for a given training scenario (including larger network architectures and types, different training sets, or training objectives), and thereby enable them to better estimate how many networks are required to ensure that the insights drawn from them will generalize. In addition to the impact on computational neuroscience, we expect the concept of representational consistency, which can be applied across different network layers, architectures, or training epochs, to also benefit machine learning researchers in understanding differences among networks operating at different levels of task performance.

## Methods

**Deep neural network training.** The main architecture used throughout all experiments presented here is All-CNN-C[7], a 9 layer fully convolutional network that exhibits the state of the art performance on the CIFAR-10 data set. To optimize architectural simplicity, the network uses only convolutional layers with a stride of 2 at layer 3 and 6 to replace max- or mean-pooling. We used the same number of feature maps (96, 96, 96, 192, 192, 192, 192, 192, 10) and kernel sizes (3, 3, 3, 3, 3, 3, 3, 1, 1) as in the original paper (Fig. 1a).

To show that our results generalize beyond a single DNN architecture we trained an additional architecture reminiscent of VGG-S[33]. In contrast to the original VGG-S architecture, we replaced the two deepest, fully connected layers with convolutional layers to reduce the number of trainable parameters and thus the training duration by ~80%. The number of feature maps used per layer was [96, 128, 256, 512, 512, 1024, 1024], and the kernel sizes were [7, 5, 3, 3, 3, 3, 3]. We used ReLU as the activation function at every layer. Mirroring the kernel sizes across layers, we refer to this architecture as "All-CNN-7".

All-CNN-C network instances were trained for 350 epochs using a momentum term of 0.9 and a batch size of 128. All networks of the All-CNN-7 architecture were trained for 250 epochs using ADAM with an epsilon term of 0.1 and a batch

size of 512. For both architectures, we used an initial learning rate of 0.01, the L2 coefficient was set to $10^{-5}$, and we performed norm-clipping of the gradients at 500. Training of the main DNNs was performed on the full CIFAR-10 image set. CIFAR-10 consists of 10 categories of objects, each of which is represented by 5000 training and 1000 test images. Ten network instances were trained for the main analyses, all without dropout. Networks were trained using Tensorflow (1.3.0) and Python 3.5.4.

In addition, we trained ten instances of AlexNet in its 2014 refined version[16] on ILSVRC 2012[17]. All training hyperparameters were chosen to match the original publication as closely as possible (learning rate 0.01, dropout 0.5, mini-batch size 128, momentum 0.9, weight decay 0.0005). Networks were trained for 90 epochs. The retrained AlexNet instances reached an average top-1 accuracy of 58.1% on ILSVRC 2012. A weighted loss was used to correct for data set imbalances in the number of images across object categories.

Network training was identical across all instances (same architecture, same data set, the same sequence of data points), with the exception of the random seed for the weight initialization. As a result, the networks only differ in the initial random weights, which are, however, sampled from the same distribution[34]. All trained neural networks are available via OSF[35].

**Representational similarity analysis and network consistency.** We characterize the internal representations of the trained networks based on representational similarity analysis (RSA)[6], a method used widely across systems neuroscience to gain insight into representations in high-dimensional spaces. An overview of the analysis pipeline is provided in Fig. 10a (Matlab 2018a was used for RSA and all other analyses presented in this manuscript).

RSA builds upon the concept of representational dissimilarity matrices (RDMs), which store all pairwise distances between the stimulus-driven pattern activations in response to a large set of input stimuli (Figs. 1a and 10a). Here we use 1000 test stimuli, 100 from each of the 10 CIFAR-10 categories, such that the resulting RDMs have a size of $1000 \times 1000$ (Figs. 1b and 10b). The RDMs are symmetric around the diagonal and therefore contain 499,500 unique distance estimates. In the current set of experiments, pairwise distances (using correlation-, cosine-, and (unit length pattern-based) Euclidean distance) are measured in the activation space of individual layers, where each unit corresponds to its own input dimension. The resulting matrix thereby characterizes the representational space spanned by the network units, as it depicts the geometric relations of all different input stimuli with respect to each other. This focus on relative distances renders RSA largely invariant to rotations of the input space (including random shuffling of input dimensions, but see Supplementary Fig. 6). It is therefore well suited for comparisons across deep neural network instances.

Because RDMs are distance matrices, they can be used as a basis for multidimensional scaling (MDS) to project the high-dimensional network activation patterns into 2D. While not a lossless operation, as high-dimensional distances can usually not be perfectly reproduced in 2D, MDS does nevertheless enable us to gain first insights into the internal organization by visualizing how network layers cluster the 1000 test images from the 10 different categories.

In addition to enabling 2D visualizations of network-internal representations (i.e., the organization of test images in high-dimensional layer activation space, Fig. 2), RDMs themselves can be used as observations (each RDM is a point in the high-dimensional space of all possible RDMs) and thereby form the basis for computing "second-level" distance matrices (Fig. 10a). The resulting distance matrices can be used to compare representations across multiple network layers and network instances (rather than test images as in first-level RDMs). Here, we compute a second-level distance matrix based on the RDMs for all network layers and instances. Again, we use MDS to visualize the data points in 2D (Fig. 3).

For a more quantitative comparison of network-internal representations, characterized here in terms of RDMs, we define representational consistency as the shared variance across representational distances observed in high-dimensional network activation space. Representational consistency is computed as a squared Pearson correlation between RDMs (Fig. 1c). If two network instances separate the test stimuli with similar geometry, the representational consistency will be high (max 1), whereas uncorrelated RDMs exhibit low representational consistency (min 0). Pseudocode for the representational consistency analysis is provided as Supplementary Information.

**Varying input statistics and representational consistency.** The main experimental manipulation in this work consists of using different random weights at the point of network initialization. To better understand the size of the effects on network-internal representations, we compared the effects observed to differences that emerge from using different images from the same categories (within-category split), or different categories altogether (across-category split). To perform this control analysis, two subsets of CIFAR-10 were created. For the across-category division, we split the training and test sets on the level of categories. This resulted in two data sets with 5 categories each while preserving the number of images per category (5000 training, 1000 test images). For the within-category division, the data set was split based on images rather than categories. This preserves the number of categories (10) but halves the number of training images per category. For an illustration of the splitting procedure that resulted in the within-category, and the across-category splits of CIFAR-10 see Supplementary Fig. 9.

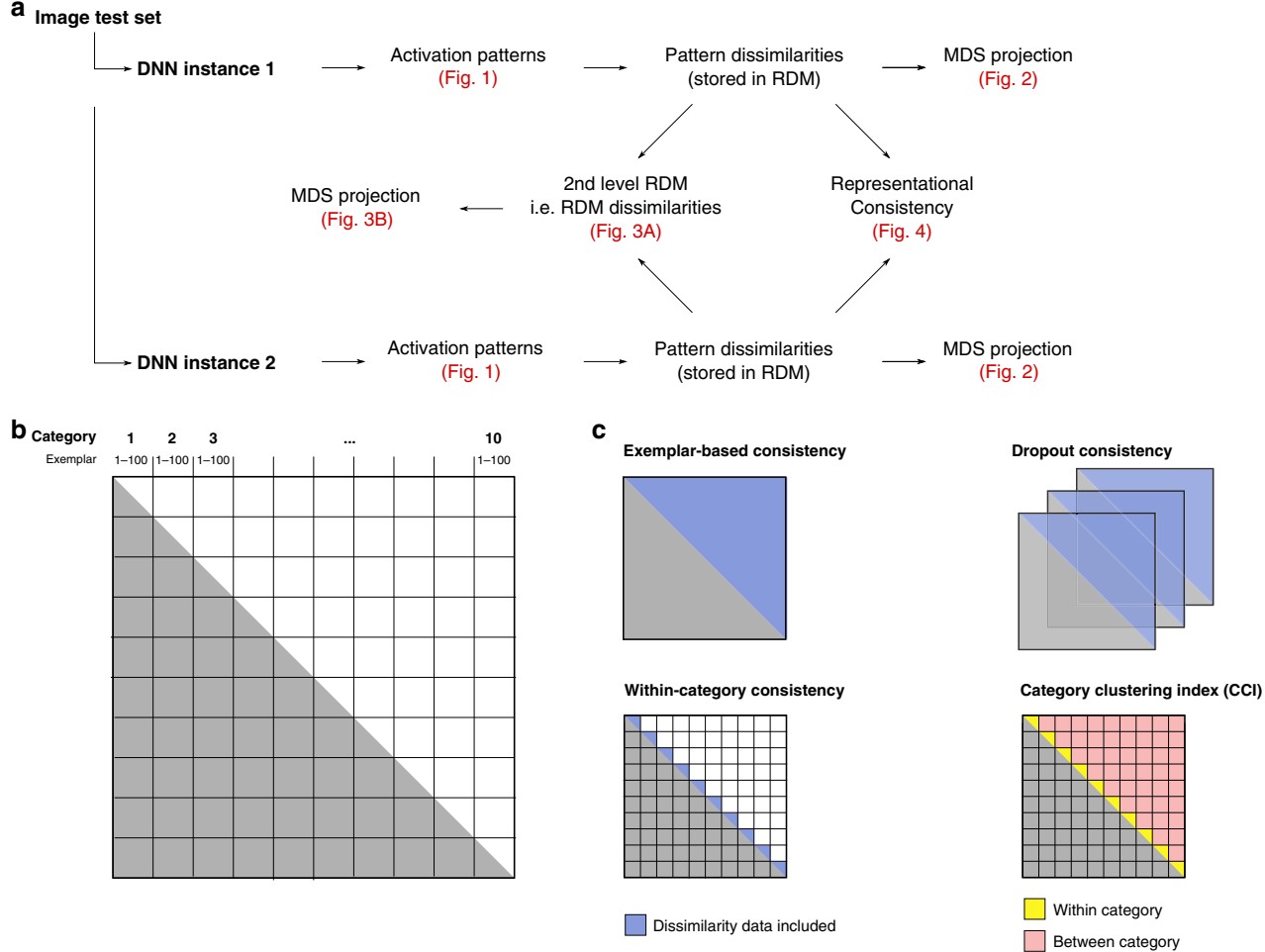

**Fig. 10 Analysis pipeline details. a** Overview of the different analysis steps taken to produce Figs. 1–4. Test images were processed by individual network instances. These activation vectors were used to compute RDMs for each network instance and layer. These distance matrices were used for MDS projection and as input to (i) representational consistency estimates, and (ii) 2nd level RSA analyses in which RDMs instead of activation patterns are compared. The second-level RDMs were projected into 2D using MDS. **b** Overview of the first-level RDM structure. These RDMs are of size 1000 × 1000, depicting the activation vector distances for 100 instances of 10 object categories. **c** Our analyses focus on different aspects of the RDM shown in **b**. Exemplar-based consistency uses all pairwise differences, whereas within-category consistency focuses on distances among exemplars of the same category only. Consistency with dropout extracts multiple RDM samples and subsequently uses their average to compute consistency. Finally, our category clustering index contrasts distances among category exemplars categories (shown in yellow) with distances between exemplars of different categories (red).

In summary, the consistency of network instances resulting from different random weight initializations (different seeds, same categories, same images), was compared with (a) different images (same seed, same categories), and (b) different categories (same seed, different images). Five networks were trained for each half of the data set for both splits (a and b, resulting in 5 × 2 = 10 network instances each). Representational consistency was computed using pairs of network instances with the same random seed (5 pairs for each split). Note that representational consistency was computed based on 1000 test images from all 10 CIFAR-10 categories, independent of the training set used to train the networks.

**Category clustering and representational consistency**. To measure how well the layers of a network separate exemplars from different categories, we computed a category clustering index (CCI), which contrasts the distances of stimuli within the same category with the distances for stimuli originating from different categories. Based on the RDM computed for the 1000 test stimuli (100 stimuli per each of 10 categories), CCI contrasts distances of category exemplars within the category with distances across categories. It is defined as CCI = (across − within)/(across + within) and was computed for each layer of each network instance trained. CCI has a maximum of 1 (all categories cluster perfectly and are perfectly separable), and a minimum of 0 (no separability, same distances across and within categories).

In addition, we investigated the relationship between CCI and representational consistency. For each layer, we computed the mean representational consistency across all 45 pairwise comparisons between 10 network instances and used a robust

Pearson correlation to demonstrate its relation to the mean class clustering indices (CCIs) across all 10 training seeds (Supplementary Fig. 4).

**Causes for decreasing representational consistency**. To better understand the origins of changes in representational consistency, we compare (i) exemplar-based consistency, (ii) centroid-based consistency, (iii) consistency of within-category distances and the (iv) effects of cocktail-blank normalization.

To understand whether a misalignment in the arrangement of individual category exemplars or the arrangement of entire classes is leading to decreased consistency, we computed the 10 class centroids and used their position in activation space to arrive at centroid-based representational consistency. This was compared with consistency based on all 1000 stimuli (exemplar-based representational consistency), and consistency computed when only distances between exemplars of the same categories were considered (within-category consistency).

To rule out effects of changed RDM size in the case of centroid-based RDMs (centroid RDMs contain 45 pairwise distances whereas the exemplar-based RDMs are composed of 499,500 entries), we computed a null distribution of RDM consistency based on centroids computed from randomly sampled classes.

Finally, to test in how far the distance measure used, rather than the representational geometries themselves, could be the source of individual differences (Supplemental Materials), we performed a cocktail-blank normalization

by subtracting the mean activation pattern across all images from each network unit, before computing the RDMs and representational consistency.

**Experiments with Bernoulli dropout**. In an additional set of experiments, we explored how network regularization, here in the form of Bernoulli dropout, can affect network-internal representations. Using the full CIFAR-10 set, we trained a set of 10 networks for each of 9 dropout levels (dropout probability ranging from 0 to 0.8, each of the resulting 90 DNNs was trained for 350 epochs). After training, we extracted network activations for a set of test images either by using no dropout at test time or by using multiple dropout samples for each test image. We obtained up to 10 samples extracted for each image while keeping the dropout mask identical across network instances and the dropout rate identical to training. We created one RDM per sample and then averaged up to 10 RDMs to obtain a single RDM representing the expected representational geometry upon dropout sampling.

**Reporting summary**. Further information on research design is available in the Nature Research Reporting Summary linked to this article.

## Data availability
Trained neural network instances for all architectures and training seeds are made available via the open science foundation (OSF) at https://osf.io/3xupm/ (DOI 10.17605/OSF.IO/3XUPM). Source data are provided with this paper.

## Code availability
The code to recreate manuscript figures is included with this paper. The code to extract activations from the trained neural network models is included in the OSF repository references above.

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

## Acknowledgements
We would like to thank Adrien Doerig and Emer Jones for their helpful comments and feedback. This project received funding from the Cambridge Trust (Vice Chancellor's Award to J.M.), German Science Foundation (DFG grant 'DynaVision' to T.C.K.), the EU's Horizon 2020 Programme (Grant Agreement No. 720270, 785907 to N.K.), and was supported by a grant from the NVIDIA GPU Program to T.C.K.

## Author contributions
Conceptualization: J.M., N.K., and T.C.K.; methodology: J.M., N.K., and T.C.K.; formal analysis: J.M. and T.C.K.; software: J.M. and C.J.S.; writing—original draft: J.M., N.K., and T.C.K.; writing—review and editing: J.M., C.J.S., N.K., and T.C.K.; funding acquisition: J.M., N.K., and T.C.K.

## Competing interests
The authors declare no competing interests.

**Additional information**

