## [Peer Review File · Nature Communications]

Reviewers' Comments:

Reviewer #1:

Remarks to the Author:

The paper explores differences between the representations, learned by networks with different initialization parameters, via representational similarity analysis. Each network is represented by a dissimilarity matrix of activations for some samples (in this case, images of everyday objects), and then a correlation between these dissimilarities is computed. The paper shows that different initialization leads to representations that are not consistent with each other. Further experiments show, among others, that this is more the case for later layers in the network, and can be reduced by regularization techniques. A recommendation is made to not treat neural network representations as fixed entities, but to examine several instances.

The topic of comparing neural network representations is interesting and timely. I think the setup of this study does a good job at investigating this idea. I have some more specific comments which in my view could be useful for the paper, but overall I think this paper is suitable for publication.

My first comment is about the organization of the paper. There are a lot of experiments with different methodological steps, which are all described in full sentences. This makes it difficult to take all the ideas in, especially if one is not as familiar with all the techniques used. I think this could be improved by (i) defining methods via equations/variables/pseudocode such that the flow of inputs/outputs can be followed and (ii) clearer organization with subheadings, bullet points, maybe moving some things to the supplementary material. Some paragraphs are currently also quite long and contain multiple ideas.

Regarding the experiments, I think the networks used might not be the most commonly used ones, but since two different networks are used, and efficiency was a factor, I think this is a reasonable choice. Perhaps it would be good to comment on whether we would expect similar behavior in other types of networks – which types did the studies comparing networks to brains use? Also, how representative is CIFAR for such studies?

On a related note, I am missing a bit what the impact is of the findings on papers that do compare networks and brains. It is clear that using a single network might not give a whole picture – but how big would the impact be on the conclusions of such a study, if a differently trained network had been used? This could be just a hypothetical example to illustrate the situation to readers outside the neuroscience community.

Some other/minor comments:

- CIFAR seems to be missing a reference
- I am not familiar with the terminology “category clustering index” and it has quite few search results – although the definition makes sense intuitively, I wonder if this is known under a different name?
- Regarding the comment of not reproducing the distances faithfully with MDS – would it be interesting to look at the stress, which points are reproduced well and which are not?

Reviewer #2:

None

Reviewer #3:

Remarks to the Author:

Overall, I think this paper addresses an interesting and important topic: the variability across individual neural networks trained with different random seeds. However, many of the results and conclusions discussed in the manuscript are closely related to prior work which has drawn similar conclusions [1, 2], though critically, [1, 2] both used variants of CCA whereas RSA is used here. Unfortunately, an in-depth discussion of the differences relative to prior work is not present, and the only justification given is that RSA is applicable to neural data (which, I will note, is also true of CCA used in prior work). There is value in corroborating prior work with a different method, but this must be made explicit. As a result, I cannot recommend acceptance in its current form. I would encourage the authors to include an in-depth discussion of the differences from prior work and address the points below in a revision.

Major comments:

- 1) There is no discussion of prior work addressing the variability across network instances in the introduction of the paper whatsoever, which is essentially the main aim of this paper. Furthermore, the discussion of prior work in the discussion is overly sparse, especially since several prior papers have made similar observations. For example, [1, 2] both observed that network representations become more dissimilar with depth (though with a variant of CCA, rather than RSA as done here). A more complete discussion of the literature is necessary to contextualize the present work. Several of the contributions made here are simply restatements of prior results shown with a different method. Of course, confirming these observations from a different perspective and method has value, but necessitates a thorough discussion of similar prior work.
- 2) All of the analyses were only performed on a single dataset - CIFAR-10. To understand the generality of these results, it would be helpful to see an additional dataset, such as TinyImageNet.
- 3) VGG-753 is an extremely confusing name, especially since the number in VGG networks typically refers to the number of layers.
- 4) In Figure 7, the observation that the representational consistency is negatively correlated with the clustering index is confounded by layer depth. The clustering of classes is required by the task and is strongly related to linear classifiability, which has been shown to increase with depth [3]. Since we also know that networks become more dissimilar with depth [1], there must be a negative correlation between representational consistency and clustering/classifiability.
- 5) For Figure 8, isn't centroid consistency effectively required by the task? Ultimately, all of these networks have to distinguish the same sets of classes in the same basis at the logit layer. As such, wouldn't the distances between class centroids be constrained to be similar? That said, it is good to confirm that this is in fact the case, but the result should be framed in the context of the expectation.
- 6) For Figure 9 and 10, why would we expect there to be rotations across inputs computed with the same set of weights? It's clear why we might want to take into account rotations in the context of the same inputs for different networks, which could have different, but aligned basis sets (as in CCA), but the basis set for different inputs on the same network is fixed, so I'm not sure why we should expect rotation.
- 7) The section on dropout shouldn't be framed broadly in terms of "network regularization." Dropout is but one member of a large and diverse set of regularization techniques. Either a) claims should be limited to the impact of dropout rather than regularization, or b) several other regularization techniques, such as weight decay, batch normalization, gradient noise, etc. should be evaluated.

8) A few comments about Figure 12:

8a) I may be misreading the plots, but the authors claim that in Figure 12, "Individual networks exhibit high consistency after the first epoch, which however decreases from thereon, indicating that task training enhances individual differences." I may be misreading the plots, but I do not see the decrease in consistency along the block diagonal of Fig 1A.

8b) Related to a, it would be helpful to include summary line plots showing the average variability at a given epoch.

8c) The pattern described -- a sharp increase in consistency in the first epoch followed by a slow rise which eventually asymptotes -- likely mirrors the learning curves of the model, and simply reflects learning. It would be helpful to include these plots, as well as the correlation between these trajectories.

Minor comments:

1) The dimension numbers in Figure 1A are too small to read easily.

2) Representational consistency is defined as using the upper triangle on p2, and as using the lower triangle on p6. The choice is arbitrary, but good to be consistent for clarity.

[1] Morcos, A., M. Raghu, and S. Bengio. 2018. "Insights on Representational Similarity in Neural Networks with Canonical Correlation." *Advances in Neural Information Processing Systems*. <http://papers.nips.cc/paper/7815-insights-on-representational-similarity-in-neural-networks-with-canonical-correlation>.

[2] Kornblith, Simon, Mohammad Norouzi, Honglak Lee, and Geoffrey Hinton. 2019. "Similarity of Neural Network Representations Revisited." *arXiv [cs.LG]*. arXiv. <http://arxiv.org/abs/1905.00414>.

[3] Alain, Guillaume, and Yoshua Bengio. 2016. "Understanding Intermediate Layers Using Linear Classifier Probes," no. 2003: 1–11.

Point by point reply

We would like to thank both reviewers for the constructive criticism and the positive overall evaluation of our submission. We appreciate the feedback, which has significantly improved our manuscript. All points raised are addressed in this resubmission as detailed in this point-by-point reply. In particular, we improved the flow of the manuscript by edits to text and figures, included a more thorough discussion of existing literature on the topic, and replicate our previous results on a more commonly used network architecture (AlexNet) trained on a large-scale object classification dataset (ILSVRC 2012). Original reviewer comments are marked **orange**, our responses are black, and changes to the manuscript are highlighted in **blue**.

Reviewer 1

The paper explores differences between the representations, learned by networks with different initialization parameters, via representational similarity analysis. Each network is represented by a dissimilarity matrix of activations for some samples (in this case, images of everyday objects), and then a correlation between these dissimilarities is computed. The paper shows that different initialization leads to representations that are not consistent with each other. Further experiments show, among others, that this is more the case for later layers in the network, and can be reduced by regularization techniques. A recommendation is made to not treat neural network representations as fixed entities, but to examine several instances.

The topic of comparing neural network representations is interesting and timely. I think the setup of this study does a good job at investigating this idea. I have some more specific comments which in my view could be useful for the paper, but overall I think this paper is suitable for publication.

My first comment is about the organization of the paper. There are a lot of experiments with different methodological steps, which are all described in full sentences. This makes it difficult to take all the ideas in, especially if one is not as familiar with all the techniques used. I think this could be improved by (i) defining methods via equations/variables/pseudocode such that the flow of inputs/outputs can be followed and (ii) clearer organization with subheadings, bullet points, maybe moving some things to the supplementary material. Some paragraphs are currently also quite long and contain multiple ideas.

Answer: We thank the reviewer for this observation and suggestions for further improvements. We have (i) reworked the manuscript to improve the overall flow, (ii) broke up paragraphs with multiple ideas, (iii) provide pseudocode for representational consistency, (iv) added a novel overview figure that introduces the overall analysis pipeline, and (v) have moved less important figures to the supplementary materials.

Pseudocode (Methods):

Algorithm 1: DNN representational consistency

input : Image test set, DNN instance 1, DNN instance 2, layer ID

output: representational consistency (shared RDM variance)

for each network instance do

for each test image do

 extract activation pattern from layer ID;

for all pairs of test images do

 compute pairwise activation pattern dissimilarity;

 store dissimilarity in RDM upper triangle;

$\rho = \text{Pearson}(RDM_{instance1}, RDM_{instance2});$

$consistency = \rho^2;$

Figure change (Methods):
Fig 11 | Analysis pipeline details. (A) Overview of the different analysis steps taken to produce Figures 1-4. Test images were processed by individual network instances. These activation vectors were used to compute RDMs for each network instance and layer. These distance matrices were used for MDS projection and as input to (i) representational consistency estimates, and (ii) 2nd level RSA analyses in which RDMs instead of activation patterns are compared. The second level RDMs were projected into 2D using MDS. (B) Overview of the first level RDM structure. These RDMs are of size 1000x1000, depicting the activation vector distances for 100 instances of 10 object categories. (C) Our analyses focus on different aspects of the RDM shown in (B). Exemplar-based consistency uses all pairwise differences, whereas within-category consistency focuses on distances among exemplars of the same category only. Consistency with dropout extracts multiple RDM samples and subsequently uses their average to compute consistency. Finally, our category clustering index contrasts distances among category exemplars categories (shown in yellow) with distances between exemplars of different categories (red).

Figure changes: Previous Figures 5, 7, and 9 were moved to the supplemental materials. They now appear as Figures S3, S4, and S5.

Regarding the experiments, I think the networks used might not be the most commonly used ones, but since two different networks are used, and efficiency was a factor, I think this is a reasonable choice. Perhaps it would be good to comment on whether we would expect similar behavior in other types of networks – which types did the studies comparing networks to brains use? Also, how representative is CIFAR for such studies?

Answer: We agree that it would be beneficial to move beyond the current ‘existence proof’ of individual differences and to expand our results to a more commonly used network architecture trained on a large-scale image dataset. We therefore trained a set of 10 AlexNet instances on ILSVRC 2012 and probed the resulting internal representations for individual differences. Replicating our previous results, we show that AlexNet, too, exhibits substantial differences in the internal network representations, most prominently in fc6, which exhibits only 62% of shared variance across representational dissimilarities.

Text change (Results): “The above results represent an important existence proof for substantial DNN individual differences that can occur in computational neuroscience analysis pipelines. To expand our experiments to network architectures commonly used to predict brain data^{3,4,14–16}, we trained and tested 10 network instances of a recent version of AlexNet¹⁷ on a large-scale object classification dataset ILSVRC 2012¹⁸. As AlexNet requires larger input images than the previously used CIFAR-10 (width/height of 224px vs 32px), we sampled a new test set that nevertheless reflects the categorical structure of CIFAR-10: 100 images from each of the 10 CIFAR-10 classes were used to compute network RDMs. Replicating our previous results, consistency was also found to decrease with increasing network depth for AlexNet. The strongest individual differences were observed in fully connected layer fc6 (62% explained variance). We observe consistency levels of 84% in the penultimate representational layer (Figure 6).”

Figure change (Results):

Fig 6 | Representational consistency declines with increasing network depth in AlexNet trained on ILSVRC 2012. We repeated our above analyses of representational consistency on a set of AlexNet instances trained on large-scale object classification dataset ILSVRC 2012. Again, we only vary the initial random seed of the network weights. In line with our previous results, we observe a decrease in representational consistency from early to late network layers. The

minimal consistency is observed in layer fc6, which exhibits 62% of shared variance across network RDMS. Please note that AlexNet requires input of size 224x224, which is significantly larger than the 32x32 image size of CIFAR-10 used earlier. Because of this, we created an independent set of larger images from the same 10 categories while following the same dataset structure (100 images per CIFAR-10 category).

On a related note, I am missing a bit what the impact is of the findings on papers that do compare networks and brains. It is clear that using a single network might not give a whole picture – but how big would the impact be on the conclusions of such a study, if a differently trained network had been used? This could be just a hypothetical example to illustrate the situation to readers outside the neuroscience community.

Answer: We have now made the impact on the neuroscience community more explicit.

Text change (Discussion): “Our finding of considerable individual differences has implications for computational neuroscience where single network instances are often used as models of information processing in the brain. If a given study compared only a single network instance to brain data, then it remains a possibility that the observation of a good (or bad) fit would be partially due to chance, as training a network off of a different random seed could have resulted in substantially different internal representations (and thereby in a different estimate of the alignment between the model and the brain). Neglecting the potentially large variability in network representations will therefore likely limit the generality of claims that can be derived from comparisons between DNNs and neural representations.”

Some other/minor comments:
CIFAR seems to be missing a reference

Answer: Corrected.

I am not familiar with the terminology “category clustering index” and it has quite few search results – although the definition makes sense intuitively, I wonder if this is known under a different name?

Answer: Thank you for pointing this out. Our category clustering index is a multivariate extension to a previously introduced “category tuning index” (McKee et al., 2014). We here chose to use the term “clustering” instead of “tuning”, as this term more directly indicates that the measure is related to the distance among category instances. We have added the missing citation to the manuscript. To make the measure more accessible, part of Figure 11 now visually depicts the distances used (within- and between-category dissimilarities) to compute CCI.

Text change (Methods): “CCI can be regarded as a multivariate extension to a previously introduced category tuning index¹⁹.”

Figure change (Methods):

Fig 11 | Analysis pipeline details. (A) Overview of the different analysis steps taken to produce Figures 1-4. Test images were processed by individual network instances. These activation vectors were used to compute RDMs for each network instance and layer. These distance matrices were used for MDS projection and as input to (i) representational consistency estimates, and (ii) 2nd level RSA analyses in which RDMs instead of activation patterns are compared. The second level RDMs were projected into 2D using MDS. (B) Overview of the first level RDM structure. These RDMs are of size 1000x1000, depicting the activation vector distances for 100 instances of 10 object categories. (C) Our analyses focus on different aspects of the RDM shown in (B). Exemplar-based consistency uses all pairwise differences, whereas within-category consistency focuses on distances among exemplars of the same category only. Consistency with dropout extracts multiple RDM samples and subsequently uses their average to compute consistency. Finally, our category clustering index contrasts distances among category exemplars categories (shown in yellow) with distances between exemplars of different categories (red).

Regarding the comment of not reproducing the distances faithfully with MDS – would it be interesting to look at the stress, which points are reproduced well and which are not?

Answer: Thank you this suggestion. We now present these data as a Supplemental Figure S1. To compute the projection accuracy for each datapoint, we parcellated the overall projection stress into the contributions of each test image.

Figure change (Supplemental):

Fig S1 | MDS stress for individual datapoint reconstructions. We computed the sum of squared deviations between the original distance estimates and the MDS reconstruction for each datapoint. In the above MDS plot, the color of each point indicates its object category, whereas the color saturation indicates the goodness of the projection (high saturation equals a good fit). Data from a given network instance across all layers and datapoints were normalized to adhere to the same color scale. As can be seen above for two network instances, the reconstruction accuracy of intermediate network layers is worse than early and late layers.

Reviewer 3

Overall, I think this paper addresses an interesting and important topic: the variability across individual neural networks trained with different random seeds. However, many of the results and conclusions discussed in the manuscript are closely related to prior work which has drawn similar conclusions [1, 2], though critically, [1, 2] both used variants of CCA whereas RSA is used here. Unfortunately, a in depth discussion of the differences relative to prior work is not present, and the only justification given is that RSA is applicable to neural data (which, I will note, is also true of CCA used in prior work). There is value in corroborating prior work with a different method, but this must be made explicit. As a result, I cannot recommend acceptance in its current form. I would encourage the authors to include an in-depth discussion of the differences from prior work and address the points below in a revision.

Major comments:

1) There is no discussion of prior work addressing the variability across network instances in the introduction of the paper whatsoever, which is essentially the main aim of this paper. Furthermore, the discussion of prior work in the discussion is overly sparse, especially since several prior papers have made similar observations. For example, [1, 2] both observed that networks representations become more dissimilar with depth (though with a variant of CCA, rather than RSA as done here). A more complete discussion of the literature is necessary to contextualize the present work. Several of the contributions made here are simply restatements of prior results shown with a different method. Of course, confirming these observations from a different perspective and method has value, but necessitates a thorough discussion of similar prior work.

Answer: Thank you for raising this important point. We added references and more detailed descriptions of these highly relevant papers to the introduction, results, and discussion sections.

Text change (Introduction): “With this, we build on and expand previous investigations of network similarities in the machine learning community. Most notably, researchers have previously employed variants of linear canonical correlation analysis (CCA) and centered-kernel alignment (CKA) to compare network internal representations. Using singular value decomposition as a pre-processing step before CCA, singular vector CCA (svCCA) was used to compare representations across networks⁶. The authors report diverging network solutions predominantly in intermediate network layers. Building on svCCA, projection-weighted-CCA (pwCCA) was introduced, which assigns different weights to CCA vectors according to their effect on the output vectors. Using this extension, the authors observed decreasing network similarities with increasing layer depth⁷. Finally, Kornblith et al. introduced centered-kernel alignment (CKA)⁸, a neuroscience inspired technique that builds upon previous CCA solutions. Using this analysis approach, the authors demonstrated that task-trained networks developed more similar representations than random networks, even when task-training was performed on different object categorization datasets. CKA furthermore identified meaningful layer correspondence between networks trained from different network initializations. This effect was strongest in early and intermediate network layers, indicating diverging network representations in later layers.”

Text change (Results): “These results are in line with previous findings demonstrating that linear class-separability increases with network depth¹³, and observations of decreasing network similarities with increasing layer depth^{6-8,21}.”

Text change (Discussion): “The observation of increased differences with increasing network depth is in line with findings from the domain of machine learning that compared network representations using methods related to CCA (svCCA⁶, pwCCA⁷, and CKA⁸).”

2) All of the analyses were only performed on a single dataset - CIFAR-10. To understand the generality of these results, it would be helpful to see an additional dataset, such as TinyImageNet.

Answer: We agree that expanding our results to larger datasets would be beneficial. To address this point, jointly with a point raised by reviewer 1, we trained 10 instances of AlexNet on ILSVRC 2012. In line with our previous results, we observe a decrease in representational consistency from early to later network layers. Consistency is lowest in layer fc6, which exhibits only 62% of shared variance across representational dissimilarities.

Text change (Results): “The above results represent an important existence proof for substantial DNN individual differences that can occur in computational neuroscience analysis pipelines. To expand our experiments to network architectures commonly used to predict brain data^{3,4,14-16}, we trained and tested 10 network instances of a recent version of AlexNet¹⁷ on a large-scale object classification dataset ILSVRC 2012¹⁸. As AlexNet requires larger input images than the previously used CIFAR-10 (width/height of 224px vs 32px), we sampled a new test set that nevertheless reflects the categorical structure of CIFAR-10: 100 images from each of the 10 CIFAR-10 classes were used to compute network RDMS. Replicating our previous results, consistency was also found to decrease with increasing network depth for AlexNet. The strongest individual differences were observed in fully connected layer fc6 (62% explained variance). We observe consistency levels of 84% in the penultimate representational layer (Figure 6).”

Figure change (Results):

Fig 6 | Representational consistency declines with increasing network depth in AlexNet trained on ILSVRC 2012. We repeated our above analyses of representational consistency on a set of AlexNet instances trained on large-scale object classification dataset ILSVRC 2012. Again, we only vary the initial random seed of the network weights. In line with our previous results, we observe a decrease in representational consistency from early to late network layers. The minimal consistency is observed in layer fc6, which exhibits 62% of shared variance across network RDMs. Please note that AlexNet requires input of size 224x224, which is significantly larger than the 32x32 image size of CIFAR-10 used earlier. Because of this, we created an independent set of larger images from the same 10 categories while following the same dataset structure (100 images per CIFAR-10 category).

3) VGG-753 is an extremely confusing name, especially since the number in VGG networks typically refers to the number of layers.

Answer: We fully agree and changed the name to ‘ConvNet8’ throughout.

4) In Figure 7, the observation that the representational consistency is negatively correlated with the clustering index is confounded by layer depth. The clustering of classes is required by the task and is strongly related to linear classifiability, which has been shown to increase with depth [3]. Since we also know that networks become more dissimilar with depth [1], there must be a negative correlation between representational consistency and clustering/classifiability.

Answer: Thank you for pointing us towards this relationship. While linear classifiability is only indirectly related to the density of the category instances around the category centroid, we agree that our results are in line with the above prediction. We explicitly discuss this relation in the results section and moved the corresponding figure to the supplemental materials.

Text change (Results): “This indicates that network layers that separate categories better exhibit stronger individual differences, as measured via nonlinear representational consistency. These results are in line with previous findings demonstrating that linear class-separability increases with

network depth¹³, and observations of decreasing network similarities with increasing layer depth^{6-8,21}”

Figure change (Results): Figure 7 was moved to the Supplement as Figure S4.

5) For Figure 8, isn't centroid consistency effectively required by the task? Ultimately, all of these networks have to distinguish the same sets of classes in the same basis at the logit layer. As such, wouldn't the distances between class centroids be constrained to be similar? That said, it is good to confirm that this is in fact the case, but the result should be framed in the context of the expectation.

Answer: We agree that linear class-separability in the penultimate layer is required for successful task performance and that the dimensionality of this layer is equal across network instances. However, this does not necessarily imply that the geometry of the categories, i.e. the arrangement of the categories in the space, is the same across network instances. Networks could show a similar level of class-separability while relying on different category geometries. This is what we test with the category centroid analysis and report, in line with the intuition of the reviewer, that the arrangement of category centroids is highly similar across network instances. We clarified this in the text.

Text change (Results): “While linear class-separability in the penultimate network layer is required for successful task completion, this does not necessarily imply centroid consistency. That is, we cannot exclude a scenario in which a pair of networks shows a similar level of class-separability, albeit a different overall arrangement of class-centroids. In this case class-separability would be high in both cases, but centroid-consistency would be low.”

Text change (Results): “Together, these results suggest that category centroids are located in similar geometric arrangements in network instances trained off of different seeds, rendering overall category placement a less likely source of the observed individual differences.”

6) For Figure 9 and 10, why would expect there to be rotations across inputs computed with the same set of weights? It's clear why we might want to take into account rotations in the context of the same inputs for different networks, which have could have different, but aligned basis sets (as in CCA), but the basis set for different inputs on the same network is fixed, so I'm not sure why we should expect rotation.

Answer: Apologies for not communicating this more clearly. We do not expect rotations to occur with the same set of weights. Rather, different network instances with different sets of weights will yield different category projections. If one projection is a rotated version of another, this should not affect the estimates of the category geometry (i.e. the RDMs). However, we show that some

distance measures, such as cosine for correlation distance, violate this expectation. We have clarified this in the text.

Text change (Results): “If different network instances learned different projections that are equivalent to a rotation in this all-positive space, then this change will not affect classification performance. However, it can affect estimates of correlation and cosine distances (see Figure S5, as well as Figure S6 demonstrating the additional effect that rotations around the origin affect correlation distances but not cosine distances).”

7) The section on dropout shouldn't be framed broadly in terms of “network regularization.” Dropout is but one member of a large and diverse set of regularization techniques. Either a) claims should be limited to the impact of dropout rather than regularization, or b) several other regularization techniques, such as weight decay, batch normalization, gradient noise, etc. should be evaluated.

Answer: We agree that Bernoulli dropout should be explicitly mentioned as the regularization technique throughout the manuscript. We have corrected this throughout the manuscript.

Text change (Results, example): “Subsequently, we explore possible causes for these individual differences and investigate their interaction with network regularization via Bernoulli dropout.”

8) A few comments about Figure 12:

8a) I may be misreading the plots, but the authors claim that in Figure 12, “Individual networks exhibit high consistency after the first epoch, which however decreases from thereon, indicating that task training enhances individual differences.” I may be misreading the plots, but I do not see the decrease in consistency along the block diagonal of Fig 1A.

Answer: Following the reviewer's suggestion in point 8b), we have added line plots to more clearly show the decrease in consistency across the training trajectory.

Figure change (Results):

Fig 10 | Final-layer representational consistency (exemplar-based) across training epochs. (A) Comparing representational consistency across early epochs [1 to 10] (left) and throughout all training epochs [1 to 350 in steps of 50] (right). Lines parallel to the main diagonal indicate that network instances remain on their distinct representational trajectory compared to other networks. **(B)** Representational consistency averaged for each training epoch demonstrates increasing individual differences with training. **(C)** Test performance across training (top-1 accuracy). **(D)** Representational consistency and test performance exhibit a strong negative relationship indicating that task training enhances individual differences.

8b) Related to a, it would be helpful to include summary line plots showing the average variability at a given epoch.

Answer: Thank you for this suggestion. We agree and added the requested line plots to the figure.

Figure change (Results): See response to 8a above.

8c) The pattern described -- a sharp increase in consistency in the first epoch followed by a slow rise which eventually asymptotes -- likely mirrors the learning curves of the model, and simply reflects learning. It would be helpful to include these plots, as well as the correlation between these trajectories.

Answer: We agree and have included line plots for mean consistency and task performance across training epochs. In line with the reviewer’s intuition, these two show a strong negative relationship, as shown in a third panel (see response to 8a right above).

Text change (Results): “In line with this, representational consistency and task performance exhibit a strong negative relationship (Figure 10B-D).”

Figure change (Results): See response to 8a.

Minor comments:

1) The dimension numbers in Figure 1A are too small to read easily.

Answer: Adjusted.

Figure change (Introduction):

2) Representational consistency is defined as using the upper triangle on p2, and as using the lower triangle on p6. The choice is arbitrary, but good to be consistent for clarity.

Answer: Corrected. Thank you for spotting the inconsistency.

Reviewers' Comments:

Reviewer #1:

Remarks to the Author:

I am happy with the changes to the manuscript and I think it is now more accessible to readers. Displaying the MDS stress with saturation is a great idea.

The only thing I would still want to point out (which I didn't notice earlier) is that I'm sad to see the statement about "providing data/code upon reasonable request". I understand that providing e.g. medical data might have privacy issues, but when many public datasets and networks are studied, sharing their (processed) versions would only increase the paper's impact and make science more inclusive.

- Veronika Cheplygina

Reviewer #3:

Remarks to the Author:

Having read the author response, I am now comfortable recommending this paper for publication. I appreciate the changes the authors have made to the manuscript and feel that the paper is now substantially stronger overall.

Point-by-point reply (reviewer)

We would like to thank the editor and the two reviewers for the very constructive overall process and of course for the positive evaluation of our work. We have addressed the last remaining comment (reviewer 1) with this submission.

Reviewer 1

I am happy with the changes to the manuscript and I think it is now more accessible to readers. Displaying the MDS stress with saturation is a great idea.

The only thing I would still want to point out (which I didn't notice earlier) is that I'm sad to see the statement about "providing data/code upon reasonable request". I understand that providing e.g. medical data might have privacy issues, but when many public datasets and networks are studied, sharing their (processed) versions would only increase the paper's impact and make science more inclusive.

- Veronika Cheplygina

Answer: We agree. Included in the submission is a .zip archive with data and code that can be used to reproduce the manuscript figures. In addition, we have uploaded the trained networks (all three DNN architectures and all seeds) to an openly available repository (OSF), together with code to extract network activations from them.

Text change:

”Data availability

Source data are provided with this paper. Trained network instances and code to extract network activations are available via the open science foundation (OSF) at: <https://osf.io/3xupm/>.

Code availability

Code to recreate manuscript figures is included with this paper. Code to extract activations from the trained neural network models is included in the OSF repository references above.”